



# Greenland Ice Sheet solid ice discharge from 1986 through 2019

Kenneth D. Mankoff[1], Anne Solgaard[1], William Colgan[1], Andreas P. Ahlstrøm[1], Shfaqat Abbas Khan[2], and Robert S. Fausto[1]

[1]Department of Glaciology and Climate, Geological Survey of Denmark and Greenland (GEUS), Copenhagen, Denmark
[2]DTU Space, National Space Institute, Department of Geodesy, Technical University of Denmark, Kgs. Lyngby, Denmark

**Correspondence:** Ken Mankoff (kdm@geus.dk)

**Abstract.** We present a 1986 through 2019 estimate of Greenland Ice Sheet ice discharge. Our data include all discharging ice that flows faster than 100 m yr$^{-1}$ and are generated through an automatic and adaptable method, as opposed to conventional hand-picked gates. We position gates near the present-year termini and estimate problematic bed topography (ice thickness) values where necessary. In addition to using annual time-varying ice thickness, our time series uses velocity maps that begin
with sparse spatial and temporal coverage and ends with near-complete spatial coverage and twelve-day updates to velocity. The 2010 through 2019 average ice discharge through the flux-gates is ~487 ±49 Gt yr$^{-1}$. The 10 % uncertainty stems primarily from uncertain ice bed location (ice thickness). We attribute the ~50 Gt yr$^{-1}$ differences among our results and previous studies to our use of updated bed topography from BedMachine v3. Discharge is approximately steady from 1986 to 2000, increases sharply from 2000 to 2005, then is approximately steady again. However, regional and glacier variability is more pronounced,
with recent decreases at most major glaciers and in all but one region offset by increases in the northwest region through 2017, and the southeast 2017 through 2019. As part of the journal's living archive option and our goal to make an operational product, all input data, code, and results from this study will be updated as needed (when new input data are available, as new features are added, or to fix bugs) and made available at doi:10.22008/promice/data/ice_discharge (Mankoff, 2019a) and at http://github.com/mankoff/ice_discharge.

## 1  What's new in this version

The data has been updated repeatedly between the first version of this paper (Mankoff et al., 2019) and this version. The data will continue to be updated, often sub-monthly, although reference papers will only come out once or twice a year. Therefore users are encouraged to regularly check for data updates at doi:10.22008/promice/data/ice_discharge when using the data.

A post-peer-review website is available at https://github.com/mankoff/ice_discharge where we document changes to the
paper and use the GitHub Issues feature to collect suggested improvements to the paper, document those improvements as they are implemented, document problems that made it through review, and mention similar papers that have been published since this was accepted. The git commit for this version of the paper is 3ff3e15 .

In this version the NSIDC 0478 ice velocity data (Joughin et al., 2015, updated 2018) have been updated from v2 to v2.1. These data are used for the baseline velocity and gate selection, and therefore gate locations have shifted slightly, and number





of gate pixels and number of gates have changed. The effect of this change is < 2.5 % of the estimated discharge. In this version there are now `5830` pixels and `268` gates.

The NSIDC 0646 ice velocity data (Howat, 2017) have been updated from v2 to v2.1. This update increases coverage and discharge in the 1980s by ~25 to 40 Gt yr$^{-1}$ (~6 to 10 %) due to higher velocity estimates than the previous product that covered

that time period with annual averages (Mouginot et al., 2018b, c). This change highlights that ice-sheet wide differences between velocity products can be non-trivial (c.f. Millan et al. (2019)). The time series has also been extended through both the updated NSIDC 0646 data and `59` Sentinel 1 velocity maps from 2018 through present (`2020-01-17`). We have also added `48` additional MEaSUREs (Joughin (2018); Joughin et al. (2010, 2018); hereafter NSIDC 0731) monthly average velocity maps from 2014-12-01 through 2018-11-30.

We have updated the time series graphics (Figures 4, 5, and 6) in the following manner: Any observation (gate-, region-, or ice-sheet wide) where coverage is < 50 % is discarded from the graphic (low coverage data is still included in the downloadable data), and annual average is only computed if there are three or more samples in a year.

Finally, the supplemental material includes significantly more meta-data about the input data used in this work to aid in both reproducibility by third-parties, and in tracking the impact of additional and updated input data on future versions of this work.

Results show a continued steady total discharge. The contributions from the central west (CW) region continue to decrease, while the central east (CE) region continues to increase, and CE and CW are now approximately tied for the 3rd largest discharging region. The top three individual contributing glaciers remain dynamic - Sermeq Kujalleq (Jakobshavn Isbræ) continued its rapid discharge decline in 2017 and 2018 returning to approximately its discharge from year 2000, until increasing again in 2019. For some time in 2018 Helheim was the top Greenlandic glacier contributing to sea level rise.

## 2 Introduction

The mass of the Greenland ice sheet is decreasing (e.g. Fettweis et al. (2017); van den Broeke et al. (2017); Wiese et al. (2016); Khan et al. (2016)). Most ice sheet mass loss – as iceberg discharge, submarine melting, and meltwater runoff – enters the fjords and coastal seas, and therefore ice sheet mass loss directly contributes to sea-level rise (WCRP Global Sea Level Budget Group, 2018; Moon et al., 2018; Nerem et al., 2018; Chen et al., 2017). Greenland's total ice loss can be estimated through

a variety of independent methods, for example 'direct' mass change estimates from GRACE (Wiese et al., 2016) or by using satellite altimetry to estimate surface elevation change, which is then converted into mass change (using a firn model, e.g. Khan et al. (2016)). However, partitioning the mass loss between ice discharge (D) and surface mass balance (SMB) remains challenging (c.f. Rignot et al. (2008) and Enderlin et al. (2014)). Correctly assessing mass loss, as well as the attribution of this loss (SMB or D) is critical to understanding the process-level response of the Greenland ice sheet to climate change, and thus

improving models of future ice-sheet changes and associated sea-level rise (Moon et al., 2018).





The total mass of an ice-sheet, or a drainage basin, changes if the mass gain (SMB inputs, primarily snowfall) is not balanced by the mass loss (D and SMB outputs, the latter generally meltwater runoff). This change is typically termed ice-sheet mass balance (MB) and the formal expression for this rate of change in mass is (e.g. Cuffey and Paterson (2010)),

$$\frac{\mathrm{d}M}{\mathrm{d}t} = \rho \int\limits_{A} b\,\mathrm{d}A - \int\limits_{g} Q\,\mathrm{d}g, \tag{1}$$

where $\rho$ is the average density of ice, $b$ is an area mass balance, and $Q$ is the discharge flux. The left hand side of the equation is the rate of change of mass, the first term on the right hand side is the area $A$ integrated surface mass balance (SMB), and the second term is the discharge $D$ mass flow rate that drains through gate $g$. Equation 1 is often simplified to

$$MB = SMB - D \tag{2}$$

where $MB$ is the mass balance, and referred to as the "input-output" method (e.g. Khan et al. (2015)). Virtually all studies
agree on the trend of Greenland mass balance, but large discrepancies persist in both the magnitude and attribution. Magnitude discrepancies include, for example, Kjeldsen et al. (2015) reporting a mass imbalance of -250 $\pm$ 21 Gt yr[-1] during 2003 to 2010, Ewert et al. (2012) reporting -181 $\pm$ 28 Gt yr[-1] during 2003 to 2008, and Rignot et al. (2008) reporting a mass imbalance of -265 $\pm$ 19 Gt yr[-1] during 2004 to 2008. Some of these differences may be due to different ice sheet area masks used in the studies. Attribution discrepancies include, for example, Enderlin et al. (2014) attributing the majority (64 %) of mass loss to
changes in SMB during the 2005 to 2009 period but Rignot et al. (2008) attributing the majority (85 %) of mass loss to changes in D during the 2004 to 2008 period.

Discharge may be calculated through several methods, including mass flow rate through gates (e.g. Enderlin et al. (2014); King et al. (2018); Mouginot et al. (2019)), or solving as a residual from independent mass balance terms (e.g. Kjær et al. (2012); Kjeldsen et al. (2015)). The gate method that we use in this study incorporates ice thickness and an estimated vertical
profile from the observed surface velocity to calculate the discharge. A typical formulation of discharge across a gate $D_g$ is,

$$D_g = \rho V H w, \tag{3}$$

where $\rho$ is the average density of ice, $V$ is depth-average gate-perpendicular velocity, $H$ is the ice thickness, and $w$ is the gate width. Uncertainties in $V$ and $H$ naturally influence the estimated discharge. At fast-flowing outlet glaciers, $V$ is typically assumed to be equal at all ice depths, and observed surface velocities can be directly translated into depth-averaged
velocities (as in Enderlin et al. (2014); King et al. (2018)). To minimize uncertainty from SMB or basal mass balance corrections downstream of a flux gate, the gate should be at the grounding line of the outlet glacier. Unfortunately, uncertainty in bed elevation (translating to ice thickness uncertainty) increases toward the grounding line.

Conventional methods of gate selection involve hand-picking gate locations, generally as linear features (e.g. Enderlin et al. (2014)) or visually approximating ice-orthogonal gates at one point in time (e.g. King et al. (2018)). Manual gate definition





is sub-optimal. For example, the largest discharging glaciers draw from an upstream radially-diffusing region that may not easily be represented by a single linear gate. Approximately flow-orthogonal curved gates may not be flow-orthogonal on the multi-decade time scale due to changing flow directions. Manual gate selection makes it difficult to update gate locations, corresponding with glacier termini retreat or advance, in a systematic and reproducible fashion. We therefore adopt an algorithmic
approach to generate gates based on a range of criteria.

Here, we present a discharge dataset based on gates selected in a reproducible fashion by a new algorithm. Relative to previous studies, we employ ice velocity observation over a longer period with higher temporal frequency and denser spatial coverage. We use ice velocity from 1986 through 2019 including twelve-day velocities for the last ~500 days of the time series, and discharge at 200 m pixel resolution capturing all ice flowing faster than 100 m yr$^{-1}$ that crosses glacier termini into fjords.

## 10   3   Input data

Historically, discharge gates were selected along well-constrained flight-lines of airborne radar data (Enderlin et al., 2014). Recent advances in ice thickness estimates through NASA Operation IceBridge (Millan et al., 2018), NASA Oceans Melting Greenland (OMG; Fenty et al. (2016)), fjord bathymetry (Tinto et al., 2015), and methods to estimate thickness from surface properties (e.g. McNabb et al. (2012); James and Carrick (2016)) have been combined into digital bed elevation models such
as BedMachine v3 (Morlighem et al., 2017b, a) or released as independent datasets (Millan et al., 2018). From these advances, digital bed elevation models have become more robust at tidewater glacier termini and grounding lines. The incorporation of flight-line ice thickness data into higher-level products that include additional methods and data means gates are no longer limited to flight-lines (e.g. King et al. (2018)).

Ice velocity data are available with increasing spatial and temporal resolution (e.g. Vijay et al. (2019)). Until recently, ice
velocity mosaics were limited to once per year during winter (Joughin et al., 2010), and they are still temporally limited, often to annual resolution, prior to 2000 (e.g. Mouginot et al. (2018b, c)). Focusing on recent times, ice-sheet wide velocity mosaics from the Sentinel 1A & 1B are now available every twelve days (http://PROMICE.org). The increased availability of satellite data has improved ice velocity maps both spatially and temporally thereby decreasing the need to rely on spatial and temporal interpolation of velocities from annual/winter mosaics (Andersen et al., 2015; King et al., 2018; Mouginot et al., 2019).
The discharge gates in this study are generated using only surface speed and an ice mask. We use the MEaSUREs Greenland Ice Sheet Velocity Map from InSAR Data, Version 2 (Joughin et al., 2010, 2015, updated 2018), hereafter termed "MEaSUREs 0478" due to the National Snow and Ice Data Center (NSIDC) date set ID number. We use the BedMachine v3 (Morlighem et al., 2017b, a) ice mask.

For ice thickness estimates, we use surface elevation from GIMP (Howat et al. (2014, 2017); NSIDC data set ID 0715),
adjusted through time with surface elevation change from Khan et al. (2016) and bed elevations from BedMachine v3 replaced by Millan et al. (2018) where available. Ice sector and region delineation is from Mouginot and Rignot (2019). Ice velocity data are obtained from a variety of products including Sentinel 1A & 1B derived by PROMICE (see Appendix), MEaSUREs 0478, MEaSUREs 0646 (Howat, 2017), Mouginot et al. (2018b), and Mouginot et al. (2018c). Official glacier names come



from Bjørk et al. (2015). Other glacier names come from Mouginot and Rignot (2019). See Table 1 for an overview of data sets used in this work.

This work uses `462` different velocity maps, biased toward post-2015 when twelve-day ice velocities become available from the Sentinel-1 satellites. The temporal distribution is ~10 maps per year from 1986 to 2013, 14 in 2014, 25 in 2015, 36 in 2016, 5 69 in 2017, 42 in 2018, and 24 in 2019.

**Table 1.** Summary of data sources used in this work.

| Property | Name used in this paper | Reference |
|---|---|---|
| Basal Topography | BedMachine | Morlighem et al. (2017b, a) |
| Basal Topography for Southeast | | Millan et al. (2018) |
| Surface Elevation | GIMP 0715 | Howat et al. (2014, 2017) |
| Surface Elevation Change | Surface Elevation Change | Khan et al. (2016); Khan (2017) |
| Baseline Velocity | MEaSUREs 0478 | Joughin et al. (2015, updated 2018) |
| Velocity | Sentinel | Appendix |
| Velocity | MEaSUREs 0646 | Howat (2017) |
| Velocity | MEaSUREs 0731 | Joughin (2018); Joughin et al. (2010, 2018) |
| Velocity | pre-2000 | Mouginot et al. (2018b, c) |
| Sectors & Regions | Sectors & Regions | Mouginot and Rignot (2019) |
| Names | | Bjørk et al. (2015); Mouginot and Rignot (2019) |

## 4 Methods

### 4.1 Terminology

We use the following terminology, most displayed in Fig. 1:

– "Pixels" are individual 200 m x 200 m raster discharge grid cells. We use the nearest neighbor when combining data sets 10 that have different grid properties.

– "Gates" are contiguous (including diagonal) clusters of pixels.

– "Sectors" are spatial areas that have 0, 1, or > 1 gate(s) plus any upstream source of ice that flows through the gate(s), and come from Mouginot and Rignot (2019).

– "Regions" are groups of sectors, also from Mouginot and Rignot (2019), and labeled by approximate geographic region.

15 – The "baseline" period is the average 2015, 2016, and 2017 winter velocity from MEaSUREs 0478.

– "Coverage" is the percentage of total, region, sector, or gate discharge observed at any given time. By definition coverage is 100 % during the baseline period. From the baseline data, the contribution to total discharge of each pixel is calculated,





and coverage is reported for all other maps that have missing observations (Fig. A2). Total estimated discharge is always reported because missing pixels are gap-filled (see "Missing and invalid data" section below).

– "Fast-flowing ice" is defined as ice that flows more than 100 m yr$^{-1}$.

– Names are reported using the official Greenlandic names from Bjørk et al. (2015) if a nearby name exists, then Mouginot and Rignot (2019) in parentheses.

Although we refer to solid ice discharge, and it is in the solid phase when it passes the gates and eventually reaches the termini, submarine melting does occur at the termini and some of the discharge enters the fjord as liquid water (Enderlin and Howat, 2013).

## 4.2 Gate location

Gates are algorithmically generated for fast-flowing ice (greater than 100 m yr$^{-1}$) close to the ice sheet terminus determined by the baseline-period data. We apply a 2D inclusive mask to the baseline data for all ice flowing faster than 100 m yr$^{-1}$. We then select the mask edge where it is near the BedMachine ice mask (not including ice shelves), which effectively provides grounding line termini. We buffer the termini 5000 m in all directions creating ovals around the termini and once again down-select to fast-flowing ice pixels. This procedure results in gates 5000 m upstream from the baseline terminus that bisect the baseline fast-flowing ice. We manually mask some land- or lake-terminating glaciers which are initially selected by the algorithm due to fast flow and mask issues.

We select a 100 m yr$^{-1}$ speed cutoff because slower ice, taking longer to reach the terminus, is more influenced by SMB. For the influence of this threshold on our results see the Discussion section and Fig. 2.

We select gates at 5000 m upstream from the baseline termini, which means that gates are likely > 5000 m from the termini further back in the historical record (Murray et al., 2015; Wood et al., 2018). The choice of a 5000 m buffer follows from the fact that it is near-terminus and thus avoids the need for (minor) SMB corrections downstream, yet is not too close to the terminus where discharge results are sensitive to the choice of distance-to-terminus value (Fig. 2), which may be indicative of bed (ice thickness) errors.

## 4.3 Thickness

We derive thickness from surface and bed elevation. We use GIMP 0715 surface elevations in all locations, and the BedMachine bed elevations in most locations, except southeast Greenland where we use the Millan et al. (2018) bed. The GIMP 0715 surface elevations are all time-stamped per pixel. We adjust the surface through time by linearly interpolating elevation changes from Khan et al. (2016), which covers the period from 1995 to 2016. We use the average of the first and last three years for earlier and later times, respectively. Finally, from the fixed bed and temporally varying surface, we calculate the time-dependent ice thickness at each gate pixel.





### 4.4 Missing or invalid data

The baseline data provides velocity at all gate locations by definition, but individual non-baseline velocity maps often have missing or invalid data. Also, thickness provided by BedMachine is clearly incorrect in some places (e.g. fast-flowing ice that is 10 m thick, Fig. 3). We define invalid data and fill in missing data as described below.

### 4.4.1 Invalid velocity

We flag invalid (outlier) velocities by treating each pixel as an individual time series, applying a 30 point rolling window, flagging values more than 2 standard deviations outside the mean, and repeating this filter three times. We also drop the 1972 to 1985 years from Mouginot et al. (2018b) because there is low coverage and extremely high variability when using our algorithm.

This outlier detection method appears to correctly flag outliers (see Mankoff et al. (2019) for un-filtered time series graphs), but likely also flags some true short-term velocity increases. The effect of this filter is a ~1% reduction in discharge most years, but more in years with high discharge – a reduction of 3.2 % in 2013, 4.3 % in 2003, and more in the 1980s when the data is noisy. Any analysis using this data and focusing on individual glaciers or short-term changes (or lack there-of) should re-evaluate the upstream data sources.

### 4.4.2 Missing velocity

We generate an ice speed time series by assigning the PROMICE, MEaSUREs 0478, MEaSUREs 0646, and pre-2000 products to their respective reported time stamps (even though these are time-span products), or to the middle of their time span when they cover a long period such as the annual maps from Mouginot et al. (2018b, c). We ignore that any individual velocity map or pixel has a time span, not a time stamp. Velocities are sampled only where there are gate pixels. Missing pixel velocities are linearly interpolated in time, except for missing data at the beginning of the time series which are back- and forward-filled with the temporally-nearest value for that pixel (Fig. A2). We do not spatially interpolate missing velocities because the spatial changes around a missing data point are most likely larger than the temporal changes. We visually represent the discharge contribution of directly observed pixels, termed coverage (Fig. A2) as time series graphs and opacity of dots and error bars in the figures. Therefore, the gap-filled discharge contribution at any given time is equal to 100 minus the coverage. Discharge is always reported as estimated total discharge even when coverage is less than 100 %.

### 4.4.3 Invalid thickness

The thickness data appear to be incorrect in some locations. For example, many locations have fast-flowing ice, but report ice thickness as 10 m or less (Fig. 3, left panel). We accept all ice thickness greater than 20 m and construct from this a thickness versus $\log_{10}$ speed relationship. For all ice thickness less than or equal to 20 m thick we adjust thickness based this relationship (Fig. 3, right panel). We selected the 20 m thickness cutoff after visually inspecting the velocity distribution (Fig. 3, left panel). This thickness adjustment adds 20 Gt yr$^{-1}$ to our baseline-period discharge estimate with no adjustment. In the Appendix and



Table A2 we discuss the discharge contribution of these adjusted pixels, and a comparison among this and other thickness adjustments.

## 4.5 Discharge

We calculate discharge per pixel using density (917 kg m$^{-3}$), filtered and filled ice speed, projection-corrected pixel width, and adjusted ice thickness derived from time-varying surface elevation and a fixed bed elevation (Eq. 3). We assume that any change in surface elevation corresponds to a change in ice thickness and thereby neglect basal uplift, erosion, and melt, which combined are orders of magnitude less than surface melting (e.g. Cowton et al. (2012); Khan et al. (2007)). We also assume depth-averaged ice velocity is equal to the surface velocity.

We calculate discharge using the gate-orthogonal velocity at each pixel and at each timestamp – all velocity estimates are gate-orthogonal at all times, regardless of gate position, orientation, or changing glacier velocity direction over time.

Annual averages are calculated by linearly interpolating to daily, then estimating annual. The difference between this method and averaging only the observed samples is ~3 % median (5 % average, and a maximum of 10 % when examining the entire ice sheet and all years in our data). It is occasionally larger at individual glaciers when a year has few widely-space samples of highly variable velocity.

### 4.5.1 Discharge Uncertainty

A longer discussion related to our and others treatments of errors and uncertainty is in the Appendix, but here we describe how we estimate the uncertainty related to the ice discharge following a simplistic approach. This yields an uncertainty of the total ice discharge of approximately 10 % throughout the time series.

At each pixel we estimate the maximum discharge, $D_{\mathrm{max}}$, from

$$D_{\mathrm{max}} = \rho\,(V + \sigma_V)\,(H + \sigma_H)\,W, \tag{4}$$

and minimum discharge, $D_{\mathrm{min}}$, from

$$D_{\mathrm{min}} = \rho\,(V - \sigma_V)\,(H - \sigma_H)\,W, \tag{5}$$

where $\rho$ is ice density, $V$ is baseline velocity, $\sigma_V$ is baseline velocity error, $H$ is ice thickness, $\sigma_H$ is ice thickness error, and $W$ is the width at each pixel. Included in the thickness term is surface elevation change through time ($\mathrm{d}H/\mathrm{d}t$). When data sets do not come with error estimates we treat the error as 0.

We use $\rho = 917$ kg m$^{-3}$ because the gates are near the terminus in the ablation zone and ice thickness estimates should not include snow or firn, although regionally ice density may be $< 917$ kg m$^{-3}$ due to crevasses. We ignore the velocity error $\sigma_V$ because the proportional thickness error ($\sigma_H/H$) is an order of magnitude larger than the proportional velocity error ($\sigma_V/V$) yet both contribute linearly to the discharge. $W$ is location-dependent due to the errors between our working map





projection (EPSG 3413) and a more accurate spheroid model of the earth surface. We adjust linear gate width by up to ~4% in the north and ~-2.5% in the south of Greenland (area errors are up to 8%). On a pixel by pixel basis we used the provided thickness uncertainty except where we modified the thickness (H < 20 m) we prescribe an uncertainty of 0.5 times the adjusted thickness. Subsequently, the uncertainty on individual glacier-, sector-, region-, or ice sheet scale is obtained by summing, but

not reducing by the square of the sums, the uncertainty related to each pixel. We are conservative with our thickness error estimates, by assuming the uncertainty range is from $D_{\min}$ to $D_{\max}$ and not reducing by the sum-of-squares of sectors or regions.

## 5   Results

### 5.1   Gates

Our gate placement algorithm generates `5830` pixels making up `268` gates, assigned to `174` ice-sheet sectors from Mouginot and Rignot (2019). Previous similar studies have used 260 gates (Mouginot et al., 2019), 230 gates (King et al., 2018), and 178 gates (Enderlin et al., 2014).

The widest gate (~47 km) is Sermersuaq (Humboldt Gletsjer), the 2nd widest (~34 km) is Sermeq Kujalleq (Jakobshavn Isbræ). 23 additional glaciers have gate lengths longer than 10 km. The minimum gate width is 3 pixels (600 m) by definition

in the algorithm.

The average unadjusted thickness gates is 401 m with a standard deviation of 258. The average thickness after adjustment is 436 m with a standard deviation of 223. A histogram of unadjusted and adjusted thickness at all gate locations is shown in Fig. 3.

### 5.2   Discharge

Our ice discharge dataset (Fig. 4) reports a total discharge of 460 ± 49 Gt in 1986, has a minimum of 428 ± 44 Gt in 1996, increases to 443 ± 44 in 2000, further to 500 ± 50 Gt/yr in 2005, after which annual discharge remains approximately steady at 483 to 505 ± ~50 Gt/yr during the 2005 to 2019 period.

At the region scale, the SE glaciers (see Fig. 1 for regions) are responsible for 136 to 164 (± 12 %) Gt yr[-1] of discharge (approximately one third of ice-sheet wide discharge) over the 1986 to 2019 period. By comparison, the predominantly land-

terminating NO, NE and SW together were also responsible for about one third of total ice-sheet discharge during this time (Fig. 5). The discharge from most regions has been approximately steady or declining for the past decade. The NW is the only region exhibiting a persistent long-term increase in discharge – From ~89 to 115 Gt yr[-1] (23 % increase) over the 1999 through 2017 period (+ ~1.4 Gt yr[-1] or + ~1.2 % yr[-1]). This 1999 to 2017 annual average increase in NW discharge offsets declining discharge from other regions, but the NW increase stopped in 2018 and discharge in the NW dropped by 5 Gt yr[-1] (4 %) in

2019. This NW decline is then offset by a SE region increase. The largest contributing region, SE, contributed a high of 164 ± 19 Gt in 2004, but dropped to ~150 ± 18 Gt yr[-1] for the past decade.





Focusing on eight major contributors at the individual sector or glacier scale (Fig. 6), Sermeq Kujalleq (Jakobshavn Isbræ) has slowed down from an annual average high of ~51 Gt yr$^{-1}$ in 2013 to ~34 Gt yr$^{-1}$ in 2018, likely due to ocean cooling (Khazendar et al., 2019). We exclude Ikertivaq from the top 8 because that gate spans multiple sectors and outlets, while the other top dischargers are each a single outlet. The 2013 to 2016 slowdown of Sermeq Kujalleq (Fig. 6) is compensated by

the many glaciers that make up the NW region (Fig. 5). The large 2017 and 2018 reduction in discharge at Sermeq Kujalleq is partially offset by a large increase in the 2nd largest contributor, Helheim Gletsjer (Fig. 6) and a small increase in the 3rd largest contributor, Kangerlussuaq (Bevan et al., 2019).

## 6    Discussion

Different ice discharge estimates among studies likely stem from three categories: 1) changes in true discharge, 2) different
input data (ice thickness and velocity), and 3) different assumptions and methods used to analyze data. Improved estimates of true discharge is the goal of this and many other studies, but changes in true discharge (category 1) can happen only when a work extends a time series into the future because historical discharge is fixed. Thus, any inter-study discrepancies in historical discharge must be due to category 2 (different data) or category 3 (different methods). Most studies use both updated data and new or different methods, but do not always provide sufficient information to disentangle the two. This is inefficient.
To more quantitatively discuss inter-study discrepancies, it is imperative to explicitly consider all three potential causes of discrepancy. Only when results are fully reproducible – meaning all necessary data and code are available (c.f. Mankoff and Tulaczyk (2017); Rezvanbehbahani et al. (2017); Mankoff et al. (2019)) – can new works confidently attribute discrepancies relative to old works. Therefore, in addition to providing new discharge estimates, we attempt to examine discrepancies among our estimates and other recent estimates. Without access to code and data from previous studies, it is challenging to take this
examination beyond a qualitative discussion.

The algorithm-generated gates we present offer some advantages over traditional hand-picked gates. Our gates are shared publicly, are generated by code that can be audited by others, and are easily adjustable within the algorithmic parameter space. This allows both sensitivity testing of gate location (Fig. 2) and allows gate positions to systematically evolve with glacier termini (not done here). The total ice discharge we estimate is ~10 % less than the total discharge of two previous estimates
(Mouginot et al., 2019; Enderlin et al., 2014), and similar to that of King et al. (2018), who attributes their discrepancy with Enderlin et al. (2014) to the latter using only summer velocities, which have higher annual average values than seasonally-comprehensive velocity products. The gate locations also differ among studies, and glaciers with baseline velocity less than 100 m yr$^{-1}$ are not included in our study due to our velocity cutoff threshold, but this should not lead to substantially different discharge estimates (Fig. 2).
Our gate selection algorithm also does not place gates in northeast Greenland at Storstrømmen, Bredebræ, or their confluence, because during the baseline period that surge glacier was in a slow phase. We do not manually add gates at these glaciers. The last surge ended in 1984 (Reeh et al., 1994; Mouginot et al., 2018a), prior to the beginning of our time series, and these glaciers are therefore not likely to contribute substantial discharge even in the early period of discharge estimates.





We instead attribute the majority of our discrepancy with Enderlin et al. (2014) to the use of differing bed topography in southeast Greenland. When we compare our top ten highest discharging glaciers in 2000 with those reported by Enderlin et al. (2014), we find that the Køge Bugt discharge reported by Enderlin et al. (2014) is ~31 Gt, but our estimate is only ~16 Gt (~17 Gt in King et al. (2018), and similar in Mouginot et al. (2019)). The Bamber et al. (2013) bed elevation dataset that likely uses the same bed data employed by Enderlin et al. (2014) has a major depression in the central Køge Bugt bed. This region of enhanced ice thicknesses is not present in the BedMachine dataset that we and King et al. (2018) and Mouginot et al. (2019) employ (Fig. B1). If the Køge Bugt gates of Enderlin et al. (2014) are in this location, then those gates overlie Bamber et al. (2013) ice thicknesses that are about twice those reported in BedMachine v3. With all other values held constant, this results in roughly twice the discharge. Although we do not know whether BedMachine or Bamber et al. (2013) is more correct, conservation of mass suggests that a substantial subglacial depression should be evident as either depressed surface elevation or velocity (Morlighem et al., 2016).

We are unable to attribute the remaining discrepancy between our discharge estimates and those by Enderlin et al. (2014). It is likely a combination of different seasonal velocity sampling (King et al., 2018), our evolving surface elevation from Khan et al. (2016), or other previously-unpublished algorithmic or data differences, of which many possibilities exist.

Our ice discharge estimates agree well with the most recently published discharge estimate (King et al. (2018), also used by Bamber et al. (2018)), except that our discharge is slightly less. We note that our uncertainty estimates include the King et al. (2018) estimates, but the opposite does not appear be true. The minor differences are likely due to different methods. King et al. (2018) use seasonally varying ice thicknesses, derived from seasonally varying surface elevations, and a Monte Carlo method to temporally interpolate missing velocity data to produce discharge estimates. In comparison, we use linear interpolation of both yearly surface elevation estimates and temporal data gaps. It is not clear whether linear or higher-order statistical approaches are best-suited for interpolation as annual cycles begin to shift, as is the case with Sermeq Kujalleq (Jakobshavn Isbræ) after 2015. There are benefits and deficiencies with both methods. Linear interpolation may alias large changes if there are no other observations nearby in time. Statistical models of past glacier behavior may not be appropriate when glacier behavior changes.

It is unlikely that discharge estimates using gates that are only approximately flow-orthogonal and time-invariant (King et al., 2018) have large errors due to this, because it is unlikely that glacier flow direction changes significantly, but our gate-orthogonal treatment may be the cause of some differences among our approach and other works. Discharge calculated using non-orthogonal methodology would overestimate true discharge.

## 7 Data availability

This work in its entirety is available at doi:10.22008/promice/data/ice_discharge (Mankoff, 2019a). The glacier-scale, sector, region, and Greenland summed ice sheet discharge dataset is available at doi:10.22008/promice/data/ice_discharge/d/v02 (Mankoff, 2019c), where it will be updated as more velocity data become available. The gates can be found at doi:10.22008/promice/data/ice_discharge/gates/v02 (Mankoff, 2019d), the code at doi:10.22008/promice/data/ice_discharge/code/v0.



0.1 (Mankoff, 2019b), and the surface elevation change at doi:10.22008/promice/data/DTU/surface_elevation_change/v1.0.0 (Khan, 2017).

## 8   Conclusions

We have presented a novel dataset of flux gates and 1986 through 2019 glacier-scale ice discharge estimate for the Greenland
ice sheet. These data are underpinned by an algorithm that both selects gates for ice flux and then computes ice discharges.

Our results are similar to the most recent discharge estimate (King et al., 2018) but begin in 1986 - although there is low coverage and few samples prior to 2000. From our discharge estimate we show that over the past ~30 years, ice sheet discharge was ~430 Gt yr$^{-1}$ prior to 2000, rose to over 500 Gt yr$^{-1}$ from 2000 to 2005, and has held roughly steady since 2005 at near 500 Gt yr$^{-1}$. However, when viewed at a region or sector scale, the system appears more dynamic with spatial and temporal
increases and decreases canceling each other out to produce the more stable ice sheet discharge. We note that there does not appear to be any dynamic connection among the regions, and any increase in one region that was offset by a decrease in another has likely been due to chance. If in coming years when changes occur the signals have matching signs, then ice sheet discharge would decrease or increase, rather than remain fairly steady.

The application of our flux-gate algorithm shows that ice-sheet wide discharge varies by ~30 Gt yr$^{-1}$ due only to gate
position, or ~40 Gt due to gate position and cutoff velocity (Fig. 2). This variance is approximately equal to the uncertainty associated with ice-sheet wide discharge estimates reported in many studies (e.g. Rignot et al. (2008); Andersen et al. (2015); Kjeldsen et al. (2015)). We highlight a major discrepancy with the ice discharge data of Enderlin et al. (2014) and we suspect this discharge discrepancy – most pronounced in southeast Greenland – is associated with the choice of digital bed elevation model, specifically a deep hole in the bed at Køge Bugt.
Transparency in data and methodology are critical to move beyond a focus of estimating discharge quantities, towards more operational mass loss products with realistic errors and uncertainty estimates. The convention of devoting a paragraph, or even page, to methods is insufficient given the complexity, pace, and importance of Greenland ice sheet research (Catania et al., 2019). Therefore the flux gates, discharge data, and the algorithm used to generate the gates, discharge, and all figures from this manuscript are available. We hope that the flux gates, data, and code we provide here is a step toward helping others both
improve their work and discover the errors in ours.

*Author contributions.* \ KDM conceived of the algorithm approach, and wrote the code. KDM , WIC, and RSF iterated over the algorithm results and methods. ASO provided the velocity data. SAK supplied the surface elevation change data. All authors contributed to the scientific discussion, writing, and editing of the manuscript.

*Competing interests.* \ The authors declare that they have no conflict of interest.





*Acknowledgements.* Funding was provided by the Programme for Monitoring of the Greenland Ice Sheet (PROMICE). Sentinel ice velocity maps were produced from Copernicus Sentinel-1 image data, processed by ESA data as part of PROMICE, and were provided by the Geological Survey of Denmark and Greenland (GEUS) at http://www.promice.org. Parts of this work were funded by the INTAROS project under the European Union's Horizon 2020 research and innovation program under grant agreement No. 727890. We thank the reviewers for their constructive input that helped improve the paper.





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



**Figures**

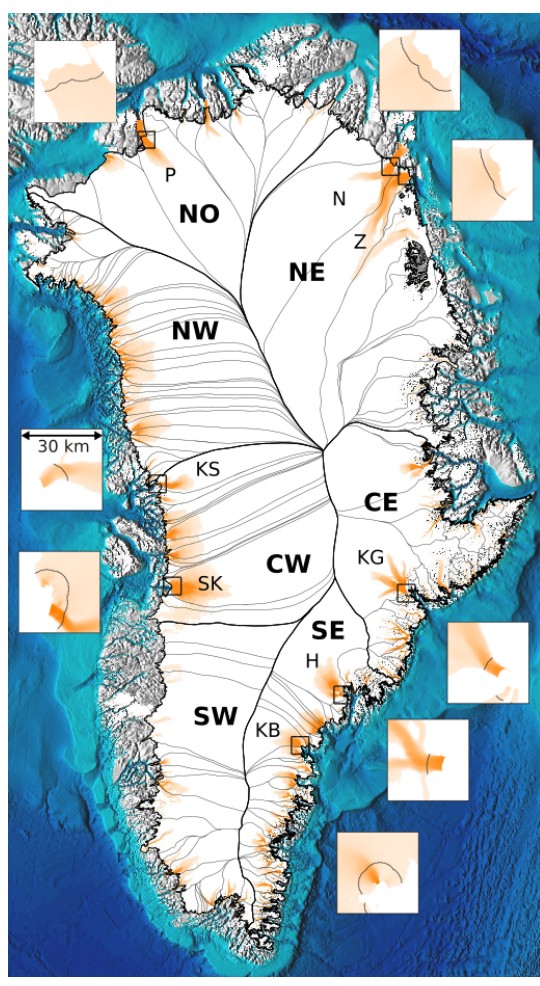

**Figure 1.** Overview showing fast-flowing ice (orange, greater than 100 m yr$^{-1}$) and the gates for eight major discharging glaciers (Fig. 6). Gates are shown as black lines in inset images. Each inset is 30 x 30 km and all have the same color scaling, but different than the main map. Insets pair with nearest label and box. On the main map, regions from Mouginot and Rignot (2019) are designated by thicker black lines and large bold labels. Sectors (same source) are delineated with thinner gray lines, and eight major discharging glaciers are labeled with smaller font. H = Helheim Gletsjer, KB = (Køge Bugt), KG = Kangerlussuaq Gletsjer, KS = Kangilliup Sermia (Rink Isbræ), N = (Nioghalvfjerdsbræ), P = Petermann Gletsjer, SK = Sermeq Kujalleq (Jakobshavn Isbræ), and Z = Zachariae Isstrøm. Basemap terrain (gray), ocean bathymetry (blues), and ice mask (white) come from BedMachine.





**Figure 2.** Heatmap and table showing ice sheet discharge as a function of gate buffer distance and ice speed cutoff. The colors of the numbers change for readability.

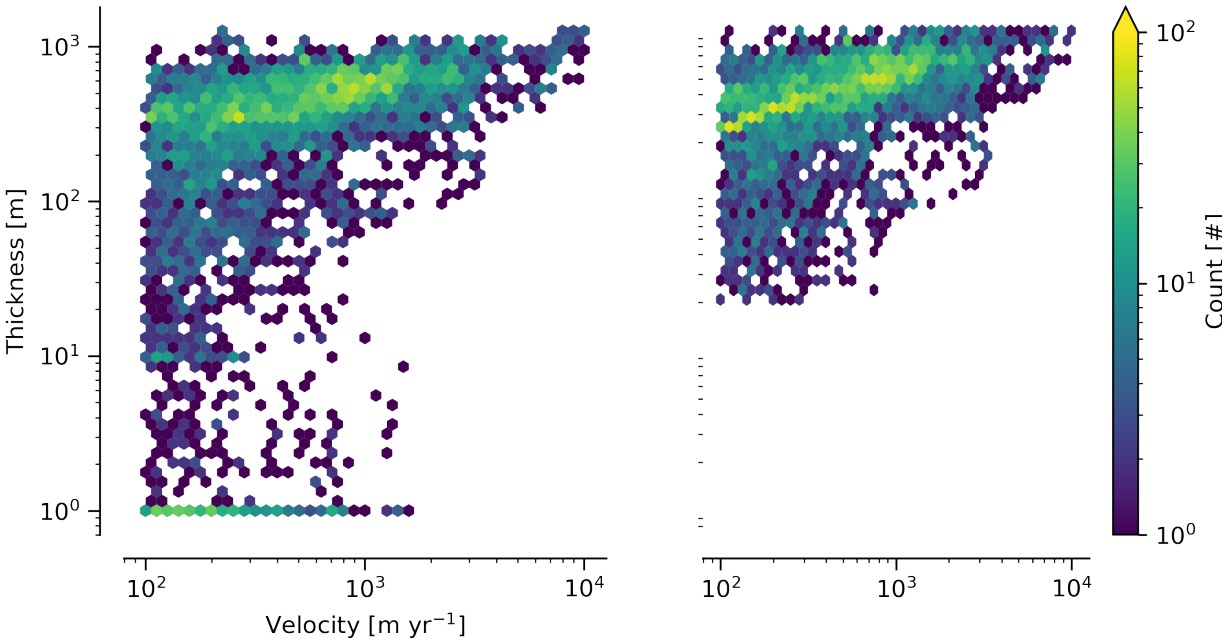

**Figure 3.** 2D histogram of velocity and thickness at all gate pixels. Left panel: Unadjusted (BedMachine & Millan et al. (2018)) thickness. Right panel: Adjusted (as described in the text) thickness.

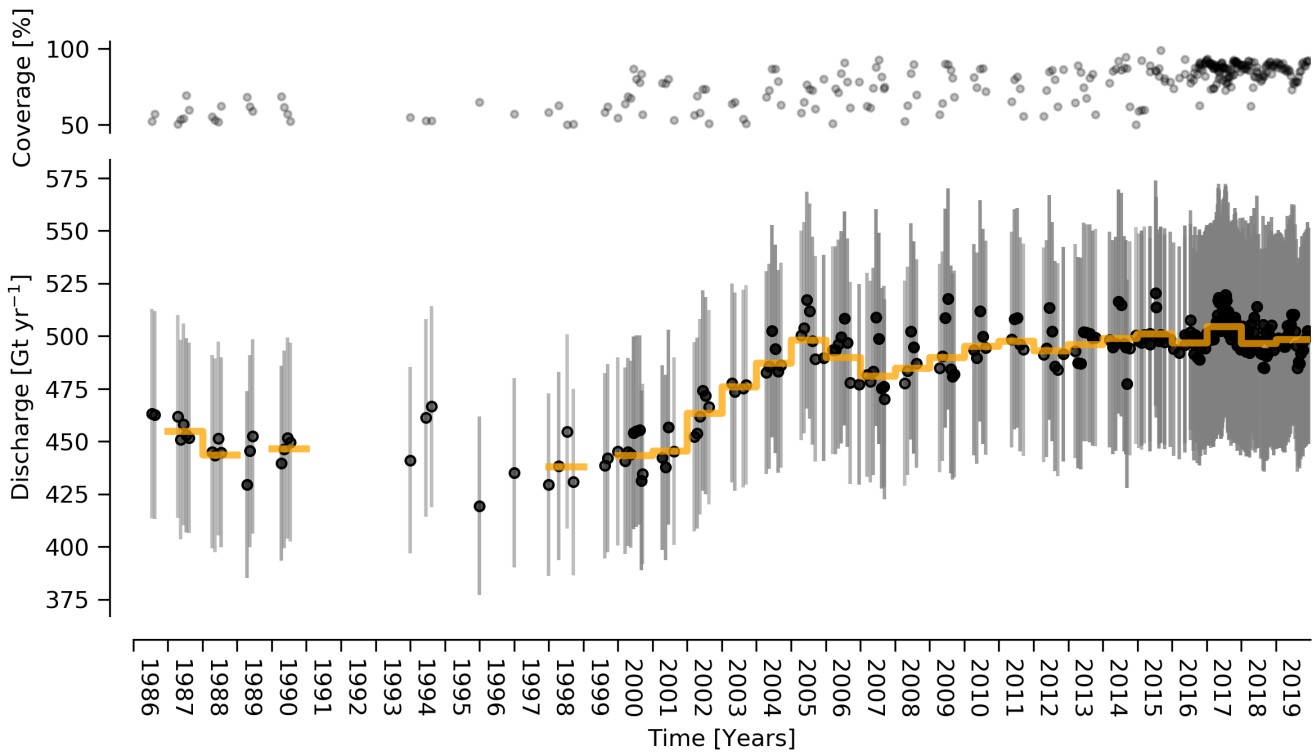

**Figure 4.** Bottom panel: Time series of ice discharge from the Greenland ice sheet. Dots represent when observations occurred (limited to coverage > 50 %). Orange stepped line is annual average (limited to three or more observations in a year). Coverage (percentage of total discharge observed at any given time) is shown in top panel, and also by opacity of dot interior and error bars on lower panel. When coverage is < 100 %, total discharge is estimated and shown.

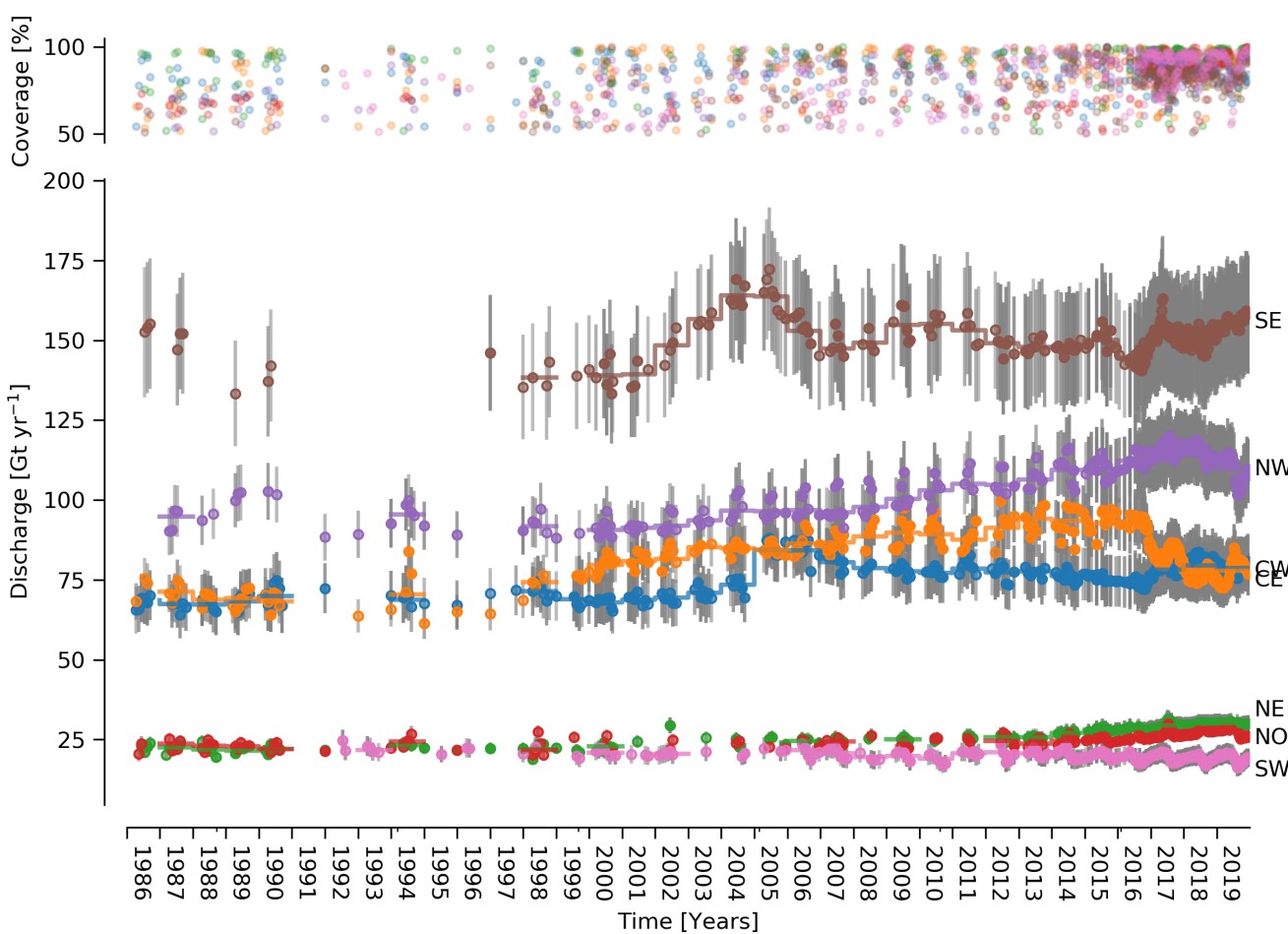

**Figure 5.** Bottom panel: Time series of ice discharge by region. Same graphical properties as Fig. 4.

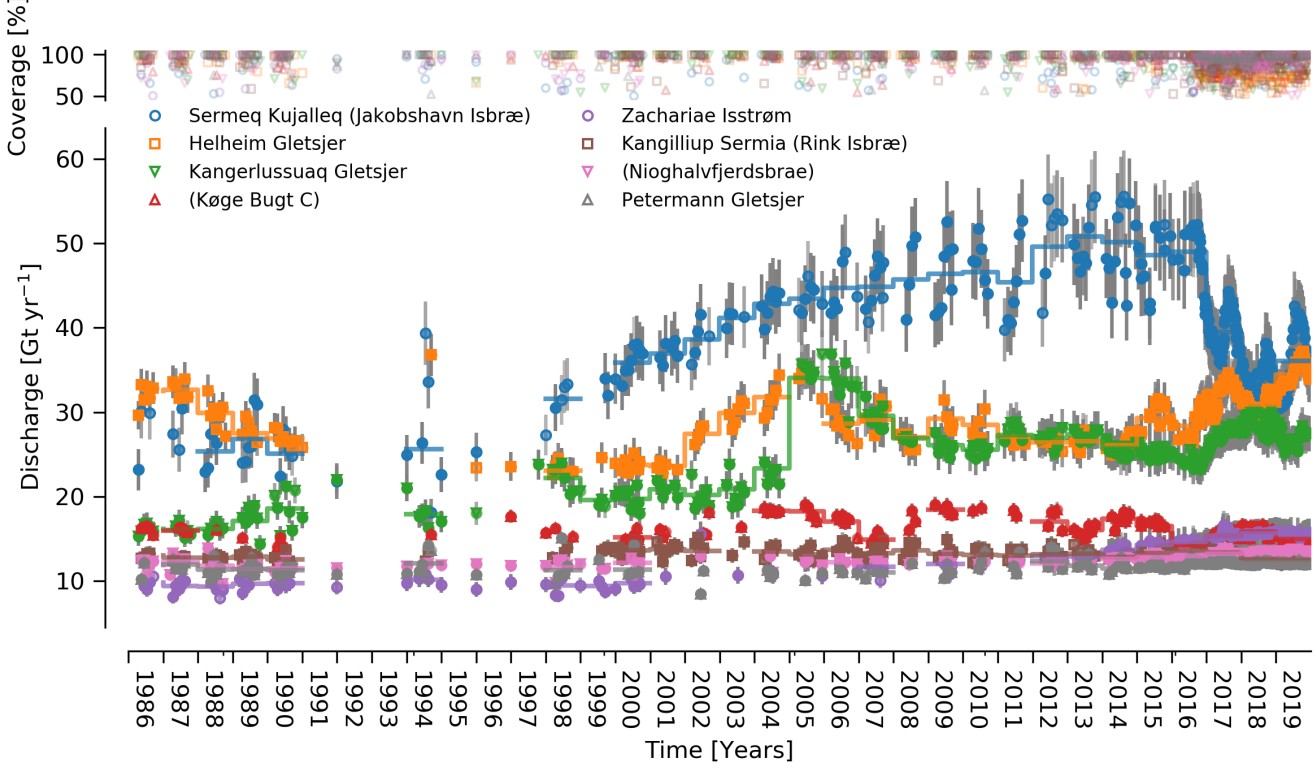

**Figure 6.** Bottom panel: Time series of ice discharge showing the eight major discharging glaciers from Figure 1. Same graphical properties as Fig. 4.



## Appendix A: Errors and Uncertainties

Here we describe our error and uncertainty treatments. We begin with a brief philosophical discussion of common uncertainty treatments, our general approach, and then the influence of various decisions made throughout our analysis, such as gate location and treatments of unknown thicknesses.

Traditional and mathematically valid uncertainty treatments divide errors into two classes: systematic (bias) and random. The primary distinction is that systematic errors do not decrease with more samples, and random errors decrease as the number of samples or measurements increases. The question is then which errors are systematic and which are random. A common treatment is to decide that errors within a region are systematic, and among regions are random. This approach has no physical basis - two glaciers a few 100 m apart but in different regions are assumed to have random errors, but two glaciers 1000s of

km apart but within the same region are assumed to have systematic errors. It is more likely the case that all glaciers less wide than some width or more deep than some depth have systematic errors even if they are on opposite sides of the ice sheet, if ice thickness is estimated with the same method (i.e. the systematic error is likely caused by the sensor and airplane, not the location of the glacier).

The decision to have $R$ random samples (where $R$ is the number of regions, usually ~18 based on Zwally et al. (2012)) is

also arbitrary. Mathematical treatment of random errors means that even if the error is 50 %, 18 measurements reduces it to only 11.79 %.

This reduction is unlikely to be physically meaningful. Our 176 sectors, 276 gates and 6002 pixels means that even if errors were 100 % for each, we could reduce it to 7.5, 6.0, or 1.3 % respectively. We note that the area error introduced by the common EPSG:3413 map projection is -5 % in the north and +8 % in the south. While this error is mentioned in some other works (e.g.

Joughin et al. (2018)) it is often not explicitly mentioned.

We do not have a solution for the issues brought up here, except to discuss them explicitly and openly so that those, and our own, error treatments are clearly presented and understood to likely contain errors themselves.

## A1   Invalid Thickness

We assume ice thicknesses < 20 m are incorrect where ice speed is > 100 m yr$^{-1}$. Of 5830 pixels, 5205 have valid thickness,

and 624 (`12 %`) have invalid thickness. However, the speed at the locations of the invalid thicknesses is generally much less (and therefore the assumed thickness is less), and the influence on discharge is less than an average pixel with valid thickness (Table ).

When aggregating by gate, there are 276 gates. Of these, 179 (67 %) have no bad pixels and 88 (33 %) have some bad pixels, 64 have > 50 % bad pixels, and 62 (23 %) are all bad pixels.

We adjust these thickness using a poor fit (correlation coefficient: 0.3) of the $\log_{10}$ of the ice speed to thickness where the relationship is known (thickness > 20 m). We set errors equal to one half the thickness (i.e. $\sigma_H = \pm 0.5\,H$). We also test the sensitivity of this treatment to simpler treatments, and have the following five categories:

**NoAdj**  No adjustments made. Assume BedMachine thickness are all correct.



**Table A1.** Statistics of pixels with and without valid thickness. Numbers represent speed [m yr⁻¹] except for the "count" row.

|  | Good Pixels | Bad Pixels |
|---|---|---|
| count | 5205 | 624 |
| mean | 857 | 272 |
| std | 1117 | 239 |
| min | 100 | 100 |
| 25% | 236 | 130 |
| 50% | 506 | 181 |
| 75% | 995 | 291 |
| max | 10044 | 1505 |

**NoAdj+Millan**  Same as NoAdj, but using Millan et al. (2018) bed where available.

**300**  If a gate has some valid pixel thicknesses, set the invalid thicknesses to the minimum of the valid thicknesses. If a gate has no valid thickness, set the thickness to 300 m.

**400**  Set all thickness < 50 m to 400 m

5  **Fit**  Use the thickness v. speed relationship described above.

Table A2 shows the estimated baseline discharge to these four treatments:

```
/home/kdm/local/anaconda/envs/sci/lib/python3.6/site-packages/numpy/core/fromnumeric.py:2495
  return ptp(axis=axis, out=out, **kwargs)
```

**Table A2.** Effect of different thickness adjustments on baseline discharge

| Treatment | Discharge (Gt) |
|---|---|
| NoAdj | 472 ± 49 |
| NoAdj+Millan | 481 ± 49 |
| 300 | 489 ± 49 |
| 400 | 495 ± 52 |
| Fit | 493 ± 51 |

Finally, Figure A1 shows the geospatial locations, concentration, and speed of gates with and without bad pixels.

10  ## A2   Missing Velocity

We estimate discharge at all pixel locations for any time when there exists any velocity product. Not every velocity product provides velocity estimates at all locations, and we fill in where there are gaps by linear interpolating velocity at each pixel in

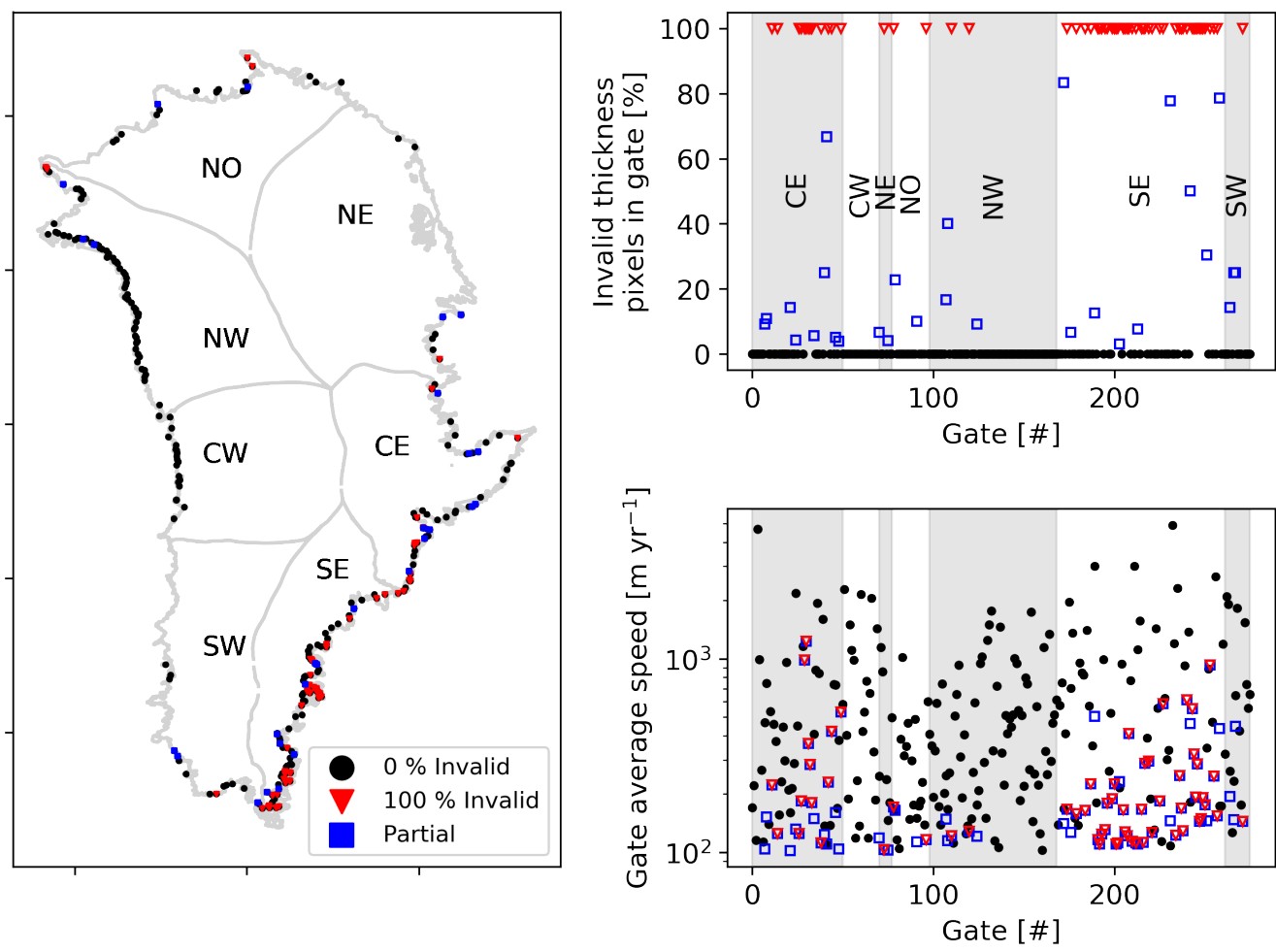

**Figure A1.** Gate locations and thickness quality. Left: locations of all gates. Black dots represent gates with 100 % valid thickness pixels, blue with partial, and red with none. Top right: Percent of bad pixels in each of the 276 gates, arranged by region. Bottom panel: Average speed of gates. Color same as left panel.

time. We calculate coverage, the discharge-weighted percent of observed velocity at any given time (Figure A2), and display coverage as 1) line plots over the time series graphs, 2) opacity of the error bars and 3) opacity of the infilling of time series dots. Linear interpolation and discharge-weighted coverage is illustrated in Figure A2, where pixel A has a velocity value at all three times, but pixel B has a filled gap at time $t_3$. The concentration of valid pixels is 0.5, but the weighted concentration, or coverage, is 9/11 or ~0.82. When displaying these three discharge values, $t_1$ and $t_4$ would have opacity of 1 (black), and $t_3$ would have opacity of 0.82 (dark gray).

This treatment is applied at the pixel level and then weight-averaged to the gate, sector, region, and ice sheet results.





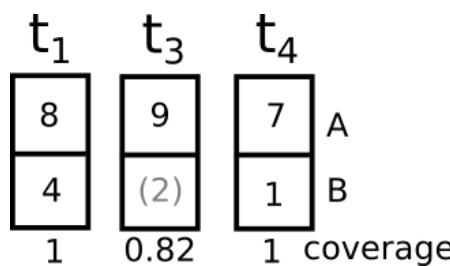

**Figure A2.** Schematic demonstrating coverage. Velocities are filled with linear interpolation in time, and coverage is weighted by discharge. $t$ columns represent the same two gate pixels (A & B) at three time steps, where $t_n$ are linearly spaced, but $t_2$ is not observed anywhere on the ice sheet and therefore not included. Numbers in boxes represents example discharge values. Gray parenthetical number is filled, not sampled, in pixel B at time $t_3$. Weighted filling computes the coverage as $9/11 = 0.\overline{81}$, instead of 0.5 (half of the pixels at time $t_3$ have observations).





**Appendix B: Køge Bugt Bed Change between Bamber et al. (2013) and Morlighem et al. (2017b)**

**Figure B1.** Differences between BedMachine (Morlighem et al., 2017b) and Bamber et al. (2013) near Køge Bugt. Panel (a) is baseline ice speed, (b) BedMachine thickness, (c) Bamber et al. (2013) thickness, and (d) difference computed as BedMachine - Bamber. Curved line is gate used in this work.



**Appendix C: Sentinel-1 ice velocity maps**

We use ESA Sentinel-1 synthetic aperture radar (SAR) data to derive ice velocity maps covering the Greenland Ice Sheet margin using offset tracking (Strozzi et al., 2002) assuming surface parallel flow using the digital elevation model from the Greenland Ice Mapping Project (GIMP DEM, NSIDC 0645) by Howat et al. (2014, 2015). The operational interferometric
5   post processing (IPP) chain (Dall et al., 2015; Kusk et al., 2018), developed at the Technical University of Denmark (DTU) Space and upgraded with offset tracking for ESA's Climate Change Initiative (CCI) Greenland project, was employed to derive the surface movement. The Sentinel-1 satellites have a repeat cycle of 12 days, and due to their constellation, each track has a twelve-day repeat cycle. We produce a Greenland wide product that spans two repeat cycles of Sentinel-1 A. The product is a mosaic of all the ice velocity maps based on 12 day pairs produced from all the tracks from Sentinel-1 A and B covering
10   Greenland during those two cycles. The product thus has a total time span of 24 days. Twelve-day pairs are also included in each mosaic from track 90, 112 and 142 covering the ice sheet margin in the south as well as other tracks on an irregular basis in order to increase the spatial resolution. Rathmann et al. (2017) and Vijay et al. (2019) have exploited the high temporal resolution of the product to investigate dynamics of glaciers. The maps are available from 2016-09-13 and onward, are updated regularly, and are available from http://promice.org.



**Appendix D: Software**

This work was performed using only open-source software, primarily `GRASS GIS` (Neteler et al., 2012) and `Python` (Van Rossum and Drake Jr, 1995), in particular the `Jupyter` (Kluyver et al., 2016), `pandas` (McKinney, 2010), `numpy` (Oliphant, 2006), `statsmodel` (Seabold and Perktold, 2010), `x-array` (Hoyer and Hamman, 2017), and `Matplotlib` (Hunter, 2007) packages. The entire work was performed in `Emacs` (Stallman, 1981) using `Org Mode` (Schulte et al., 2012). The `parallel` (Tange, 2011) tool was used to speed up processing. We used `proj4` (PROJ contributors, 2018) to compute the errors in the EPSG 3413 projection. All code used in this work is available in the Supplemental Material.