# Peer review of "Greenland Ice Sheet solid ice discharge from 1986 through March 2020"

_Earth System Science Data, 2019_

## Referee Comment (RC1) · Anonymous Referee #1 · 17 Feb 2020

Review of
**Greenland Ice Sheet solid ice discharge from 1986 through 2019**
*by Mankoff et al., 2020.*

The manuscript describes updates to the previously published Greenland-wide ice discharge data paper, including an extension of the time series through 2019, the integration of updated velocity products, and the inclusion of a designated GitHub repository. The updated paper also includes some brief descriptions of changing regional behavior, including the interesting observation that Helheim Glacier (southeast) was briefly the largest-contributing glacier as Jakobshavn Isbræ continues to decelerate. Finally, the authors have refined the discharge time series by excluding data points with less than 50% coverage. The data, methods, and meta-data are transparent and accessible, and the manuscript and discharge data product are thus valuable to the glaciology community.

I have reviewed the updates from Mankoff et al. (2019) and include several minor comments below. The manuscript is publishable once these comments are addressed.

4.4.2: I suggest also adding a statement here that only datapoints with >50% coverage are reported.

Conclusions:
Update statistics in second paragraph beginning with "ice sheet discharge was ~430 G/yr prior to 2000"
It does not seem this is accurate now with the updated velocity. More like ~440 Gt/yr?

Supplement:
Make sure all statistics and numbers are updated, including:

Appendix A: Errors and uncertainties
Fourth paragraph: replace "276 gates and 6002 pixels" with 268 gates and 5830 pixels.

A1 invalid thickness: 5205 valid + 624 invalid = 5829, not 5830.

Second paragraph of A1 and in Figure 1A caption: replace "276 gates" with 268.

---

## Referee Comment (RC2) · Anonymous Referee #2 · 18 Feb 2020

Summary: The manuscript has been modified from a previous version to account for updates to velocity datasets used for the analysis and their impacts on results. The updates are quite minor and I recommend only a few minor technical corrections below.

Technical Corrections: -It would be helpful to add a sentence explaining why the updated datasets result in a decrease in the number of pixels (and glaciers, and regions) included in the analysis. It's unclear why presumably improved velocity data would decrease the number of velocity pixels used in the flux calculations.

-Was the change from 6 day coverage to 12 day coverage a change in the dataset or was this an error that had previously made it through review?

-In reading the tracked version of the manuscript, there were a few places where the

text was a little difficult to read. Most of these places would benefit from a comma. In others the wording is awkward. For example, in the abstract "and the southeast 2017 through 2019" seems odd. Please go through the text and check the sentence structure where you've made revisions.

---

## Author Response (AR1)

**Reply to Reviewers**

**Ken Mankoff *et al.**

Comments from reviewers are in normal font and differentiated from the replies that use a bold colored font.

**1 Review 1**

The manuscript describes updates to the previously published Greenland-wide ice discharge data paper, including an extension of the time series through 2019, the integration of updated velocity products, and the inclusion of a designated GitHub repository. The updated paper also includes some brief descriptions of changing regional behavior, including the interesting observation that Helheim Glacier (southeast) was briefly the largest-contributing glacier as Jakobshavn Isbræ continues to decelerate. Finally, the authors have refined the discharge time series by excluding data points with less than 50% coverage. The data, methods, and meta-data are transparent and accessible, and the manuscript and discharge data product are thus valuable to the glaciology community.

I have reviewed the updates from Mankoff *et al.* (2019) and include several minor comments below. The manuscript is publishable once these comments are addressed.

4.4.2: I suggest also adding a statement here that only datapoints with >50% coverage are reported.

**Done.**

Conclusions:

Update statistics in second paragraph beginning with "ice sheet discharge was ~430 G/yr prior to 2000"

It does not seem this is accurate now with the updated velocity. More like ~440 Gt/yr?

**Fixed.**

Supplement:

Make sure all statistics and numbers are updated, including:

Appendix A: Errors and uncertainties

Fourth paragraph: replace "276 gates and 6002 pixels" with 268 gates and 5830 pixels.

**Fixed, and we found a bug. The count included the header line. sectors, gates, and pixels have all been reduced by one to 173, 267, and 5829 respectively.**

A1 invalid thickness: 5205 valid + 624 invalid = 5829, not 5830.

**You caught the bug too.**

Second paragraph of A1 and in Figure 1A caption: replace "276 gates" with 268.

**Now 267.**

**2    Review 2**

Summary: The manuscript has been modified from a previous version to account for updates to velocity datasets used for the analysis and their impacts on results. The updates are quite minor and I recommend only a few minor technical corrections below.

Technical Corrections: -It would be helpful to add a sentence explaining why the updated datasets result in a decrease in the number of pixels (and glaciers, and regions) included in the analysis. It's unclear why presumably improved velocity data would decrease the number of velocity pixels used in the flux calculations.

**Done in the "What's New" section.**

Was the change from 6 day coverage to 12 day coverage a change in the dataset or was this an error that had previously made it through review?

**We would classify it as an error that previously made it through review, but "error" may be too strong a word. The description in Appendix C has been updated and "six" replaced with "twelve" but there is still a six-day component, we are now just no longer highlighting it, and highlight the 12-day component.**

**In a bit more detail, the velocity product spans 24 days. But it is a sliding window of two (2) Greenland-wide coverages and updated every 12 days, with 12 day overlaps to the previous and next velocity product. The Greenland-wide coverage that takes 12 days is actually two six-day product, because Sentinel-1 A and Sentinel-1 B are offset by 6 days.**

In reading the tracked version of the manuscript, there were a few places where the text was a little difficult to read. Most of these places would benefit from a comma. In others the wording is awkward. For example, in the abstract "and the southeast 2017 through 2019" seems odd. Please go through the text and check the sentence structure where you've made revisions.

**We have tried to clarify the text and improve readability.**

[revised manuscript text omitted]